# Estimation of Respiratory States Based on a Measurement Model of Airflow Characteristics in Powered Air-Purifying Respirators Using Differential Pressure and Pulse Width Modulation Control Signals—In the Development of a Public-Oriented Powered Air-Purifying Respirator as an Alternative to Lockdown Measures

**DOI:** 10.3390/s25092939

**Published:** 2025-05-07

**Authors:** Yusaku Fujii, Akihiro Takita, Seiji Hashimoto, Kenji Amagai

**Affiliations:** School of Science and Technology, Gunma University, Kiryu 376-8515, Gunma, Japan; takita@gunma-u.ac.jp (A.T.); hashimotos@gunma-u.ac.jp (S.H.); amagai@gunma-u.ac.jp (K.A.)

**Keywords:** Powered Air-Purifying Respirator, PAPR, supply flow rate, exhaust flow rate, differential pressure, PWM duty value, respiratory state estimation, respiratory airflow simulator, inverse pressure control, wearing comfort, public-oriented PAPR

## Abstract

**Highlights:**

**What are the main findings?**
A method was developed to estimate respiratory states (exhalation/inhalation) in a public-oriented PAPR based on differential pressure and PWM duty value;The estimation model, derived under static conditions, was shown to remain valid even under dynamic conditions where the pump is actively controlled.

**What is the implication of the main finding?**
The ability to infer respiratory states enables pressure control that assists breathing by reducing pressure during exhalation and increasing it during inhalation;This respiratory-assist control has the potential to enhance wearing comfort, contributing to the feasibility of PAPRs as alternatives to lockdowns for airborne infection control.

**Abstract:**

Fluid dynamics modeling was conducted for the supply unit of a Powered Air-Purifying Respirator (PAPR) consisting of a nonwoven fabric filter and a pump, as well as for the exhaust filter (nonwoven fabric). The supply flow rate ***Q*_1_** was modeled as a function of the differential pressure **Δ*P*** and the duty value **d** of the PWM control under a constant pump voltage of ***V*** = 12.0 [V]. In contrast, the exhaust flow rate ***Q*_2_** was modeled solely as a function of **Δ*P***. To simulate the pressurized hood compartment of the PAPR, a pressure buffer and a connected “respiratory airflow simulator” (a piston–cylinder mechanism) were developed. The supply unit and exhaust filter were connected to this pressure buffer, and simulated respiratory flow was introduced as an external disturbance flow. Under these conditions, it was demonstrated that the respiratory state—i.e., the expiratory state (flow from the simulator to the pressure buffer) and the inspiratory state (flow from the pressure buffer to the simulator)—can be estimated from the differential pressure **Δ*P***, the pump voltage ***V***, and the PWM duty value **d**, with respect to the disturbance flow generated by the respiratory airflow simulator. It was also confirmed that such respiratory state estimation remains valid even when the duty value **d** of the pump is being actively modulated to control the internal pressure of the PAPR hood. Furthermore, based on the estimated respiratory states, a theoretical investigation was conducted on constant pressure control inside the PAPR and on the inverse pressure control aimed at supporting respiratory activity—namely, pressure control that assists breathing by depressurizing when expiratory motion is detected and pressurizing when inspiratory motion is detected. This study was conducted as part of a research and development project on public-oriented PAPR systems, which are being explored as alternatives to lockdown measures in response to airborne infectious diseases such as COVID-19. The present work specifically focused on improving the wearing comfort of the PAPR.

## 1. Introduction

In this study, we investigate a low-cost Powered Air-Purifying Respirator (PAPR) [1,2], designed for public use as an alternative to lockdown measures. We develop fluid dynamics models for the supply unit and exhaust filter and attempt to estimate respiratory flow based on PAPR operational parameters such as differential pressure and motor output (applied voltage and the duty value of the PWM control). If respiratory flow can be estimated with sufficient accuracy from these operational parameters, it becomes possible to implement pressure control that assists respiration—namely, depressurizing the hood during exhalation and pressurizing it during inhalation. This would enable the realization of a PAPR that makes breathing easier while being worn. In this paper, the terms exhalation and inhalation are primarily used to describe the phases of the respiratory cycle. These correspond to expiration and inspiration, respectively, which are also commonly used in physiological and engineering contexts.

The COVID-19 pandemic, which began in early 2020, had a profound impact on society [3,4,5]. Future pandemics are likely to occur due to new variants or emerging airborne infectious diseases [6,7]. While preventive measures such as mask-wearing, ventilation, and vaccination have been implemented, lockdowns [8] are sometimes introduced when responses are delayed, resulting in significant social and economic burdens [9,10]. Thus, it is crucial to develop strong and sustainable infection-control measures that can serve as alternatives to lockdowns.

Airborne transmission—alongside contact, oral, and droplet transmission—is a major route of infection. Compared to other transmission routes, airborne transmission is more difficult to control due to the wide dispersion of aerosols in the air [11,12], and is considered a likely primary route in the next pandemic. At present, apart from vaccination, infection-control options available to the general public are limited and often insufficient [13,14].

Medical-grade PAPRs are widely used as protective equipment against airborne infectious diseases. First, PAPRs are equipped with high-performance nonwoven fabric filters such as HEPA filters [15], which can capture over 99.97% of particles at the most penetrating particle size (MPPS), typically around 0.3 μm [16,17]. In general, there is a trade-off between aerosol filtration efficiency and airflow resistance in nonwoven filters; however, new types of filters based on novel principles that overcome this limitation have also been developed [18,19].

Second, PAPRs maintain positive pressure inside the hood, preventing aerosol infiltration from the outside. As a result, the Assigned Protection Factor (APF) of helmet/hood-type PAPRs reaches 1000 [20,21], significantly surpassing that of half-mask respirators (APF = 10) and N95 masks. In contrast, N95 masks exhibit negative pressure during inhalation, structurally allowing outside air to enter through gaps.

Third, the use of PAPRs can dramatically reduce the amount of inhaled virus, increasing the likelihood of remaining below the infection threshold for COVID-19 (300–2000 virions) [22,23], thereby reducing infection risk.

We have been developing a low-cost PAPR with a simplified structure while maintaining performance equivalent to medical-grade devices and exploring technical and societal challenges for practical deployment. This work is motivated by three key objectives. First, it diversifies infection-control strategies by introducing a new countermeasure for airborne diseases in addition to vaccination and lifestyle changes. Second, widespread adoption of PAPRs could provide a practical alternative to lockdowns and reduce associated social and economic losses. Third, the design and operation model of an affordable, publicly usable PAPR may serve as a sustainable foundation for future pandemic preparedness. Through these efforts, this study aims to offer a new option for airborne disease mitigation and contribute to strengthening societal resilience.

Table 1 presents the specifications of a commercial PAPR by 3M [24] and our low-cost prototype [25]. The protective performance of respiratory protective equipment is evaluated using the Assigned Protection Factor (APF), defined by NIOSH (National Institute for Occupational Safety and Health) as the ratio of outside to inside concentration: APF = [external concentration]/[internal concentration]. While medical face masks typically have an APF of 10, the 3M PAPR achieves an APF of 1000 [26] owing to its positive pressure design that physically prevents inflow of external air through any facial gaps. Additionally, the use of nonwoven fabric filters ensures the physical removal of virus-containing particles, making the system effective regardless of virus strain.

The prototype PAPR [25] developed in this study incorporates a HEPA filter (cut from a commercially available air purifier filter), a pump, and a battery, with a total component procurement cost of approximately USD 40. The prototype was assembled at Cebu Technological University (Cebu City, Philippines), and the components were procured through online shopping sites (such as Lazada). As a result, the cost of components was kept low. The device supplies filtered air—purified via a HEPA filter capable of removing over 99.97% of 0.3 μm particles—into the hood to maintain a positive pressure environment. Consequently, even if small gaps exist between the hood and the user’s face, outside air inflow can be physically prevented. Exhaust is released passively through the exhaust filter via differential pressure, which may offer some virus emission suppression if the wearer is infected. This design achieves the core functionality of a commercial medical-grade PAPR.

Commercial medical PAPRs (e.g., 3M [24] and CleanSpace [27,28]) generally consist of nonwoven fabric filters, pumps, and batteries in a relatively simple structure, yet they are often priced above USD 1000. This high cost is likely due to their intended use in high-risk healthcare environments and the lack of mass production. During the COVID-19 pandemic, the low-cost fabrication of PAPR-equivalent devices using 3D printing was also proposed [29].

The flow rates of the PAPRs shown in Table 1 range from 180 to 200 [L/min] for commercially available units, and 400 [L/min] for the prototype developed by the authors—values significantly higher than the resting respiratory flow rate of an adult male (approximately 6 L/min) [30]. This high flow rate is intended to prevent the accumulation of exhaled carbon dioxide within the PAPR hood. With such high air supply rates, these PAPRs achieve relatively comfortable wearing conditions without fogging the face shield due to exhaled moisture. However, such high flow rates can also be considered a substantial waste, suggesting that the flow path design is critical to ensure the rapid expulsion of exhaled air from the hood.

We previously evaluated the potential of PAPRs as an alternative to lockdown and as an infection-control measure using a simplified simulation assuming airborne and droplet transmission as primary routes [2]. The results showed that, with a PAPR that achieves 100% supply side filtration efficiency, continuous wearing by 60% of the population is sufficient to shift from a rapid spread scenario with an effective reproduction number ***R*_t_** = 2 to a suppression scenario with ***R*_t_target_** = 0.8. Conversely, if the wearing rate is 100% (i.e., the entire population wears the device at all times), a reduction in infection probability (***I*_r_in_**) by 60% would yield a similar effect. Assuming the cycle of generational transmission is half a month, the number of infected individuals increases fourfold every month under ***R*_t_** = 2 (since 2^2^ = 4), resulting in explosive growth. In contrast, if ***R*_t_** is reduced to ***R*_t_target_** = 0.8, the number of infections decreases to 0.64 times per month (0.8^2^ = 0.64), indicating a transition to a controlled state.

To ensure public acceptance of infection-control technologies, not only effectiveness and safety but also transparency, equity, adaptability, and privacy considerations are essential. For daily PAPR use, significant improvements in comfort, usability, and design are required, and innovation from both industry and government is expected.

As an attempt to enhance wearing comfort, we have proposed a fluid model that enables real-time pressure control in response to respiratory activity [2]. This model estimates breathing motion from supply and exhaust flow rates based on differential pressure and pump voltage and realizes assisted breathing by increasing positive pressure during inhalation and reducing it during exhalation. Additionally, the model enables the detection of improper fitting through leakage flow estimation, cough detection via pressure fluctuations, and infection estimation by integrating body temperature information. These ideas are disclosed in patents [31,32].

Existing technologies related to fan control in PAPRs include methods that dynamically adjust air supply flow according to respiratory flow, aiming to reduce airflow and extend filter lifespan [33]. For example, “The air supply flow is changed by adjusting the speed of the variable-frequency centrifugal fan according to the respiratory airflow states (exhalation or inhalation) detected at the half mask outlet”.

Network connectivity and integration with information systems are also important for PAPRs. We have developed a PAPR equipped with a controller that includes Bluetooth connectivity [34]. In the future, advanced pressure control synchronized with breathing activity and integration with network systems for recording and verifying wearing rates may allow us to reconcile individual freedom with effective infection control [1,2].

In this study, we experimentally derive a fluid dynamics model—i.e., a calibration formula that expresses the flow rate ***Q*** of the supply and exhaust units as a function of pump duty and differential pressure **Δ*P***—with the aim of enabling real-time pressure control synchronized with respiratory activity. This pressure control assists breathing by shifting the internal pressure inside the hood to a lower level during exhalation and to a higher level during inhalation, based on the detection of respiratory phases. Furthermore, we attempt to detect respiratory activity (i.e., exhalation and inhalation) based on the derived model.

## 2. System Configuration and Measurement Method

Figure 1 illustrates the configuration of the experimental apparatus used to investigate the operating conditions of the supply and exhaust units developed for the prototype Powered Air-Purifying Respirator (PAPR), along with their relationship to respiratory airflow. Figure 2 presents a photograph of the experimental setup, and Figure 3 shows another prototype PAPR (final version) with a structure similar to the one planned for future development, placed within the experimental setup alongside the supply and exhaust units developed and tested in this study.

The experimental apparatus was designed to simulate the internal environment of the pressurized hood of a PAPR. It consists of a pressure buffer to which the supply unit and exhaust unit are attached, and a “respiratory airflow simulator” that mimics the respiratory flow of a wearer is connected.

The pressure buffer is composed of an acrylic box mounted below a 25-mm-thick aluminum base. The acrylic box has wall thicknesses of 5 mm and internal dimensions of 290 mm in length, 140 mm in width, and 40 mm in height, corresponding to an internal volume of approximately 1.6 L. In addition, the volume of the acrylic cylinder connecting the pump outlet of the supply unit to the base (approximately 0.39 L) and the volume of the mounting fixture for the exhaust filter (approximately 0.02 L) bring the total effective internal volume of the pressure buffer to approximately 2.0 L.

The supply unit consists of an airtight chamber housing an electric pump and a HEPA filter. The electric pump used is a Sirocco Fan (model: 109BM12GC2-1, manufacturer: Sanyo Denki (Tokyo, Japan), rated voltage: 12 V, rated current: 0.6 A, maximum flow rate: 820 L/min, and maximum static pressure: 281 Pa). The HEPA filter was cut into a rectangular shape (width: 125 mm and height: 105 mm) from a replacement filter for air purifiers (model: PMMS-DCHF, manufacturer: IRIS OHYAMA, Sendai, Japan), and its expanded nonwoven surface area is approximately 0.14 m^2^. An acrylic cylinder is attached to the pump outlet, through which the filtered air is delivered into the pressure buffer.

The supply unit is constructed using the shell and adjustable harness of a lightweight work helmet as the base structure, onto which an airtight chamber is formed using polystyrene panels. Ambient air is drawn into the airtight room by the pressure differential generated across the electric pump, purified through the HEPA filter, and then supplied to the pressure buffer, which simulates the internal environment of a PAPR hood. The pressure on the pump’s inlet side (within the airtight room) is lower than the atmospheric pressure, whereas the outlet side (inside the pressure buffer) is higher than the atmospheric pressure.

For the exhaust unit, a thin nonwoven fabric, cut from a disposable cap for helmet inners (model: THDC-120, manufacturer: TRUSCO, Tokyo, Japan), is used as the exhaust filter. This fabric is adhered to a circular plastic plate with an opening diameter of approximately 29 mm. The plate is clamped between aluminum fixtures and mounted onto the base. In the PAPR prototype, this nonwoven exhaust filter is installed on the PAPR hood, enabling passive exhaust driven by the pressure difference between the inside and outside of the hood.

The “respiratory airflow simulator” consists of a cylinder–piston mechanism comprising an acrylic cylinder (inner diameter: 90 mm) and a piston fitted with a rubber O-ring mounted on an acrylic disc. The stroke length of the piston is approximately 275 mm, resulting in an effective volume of about 1.75 L. The piston is driven manually by the experimenter via a connected rod.

The differential pressure **Δ*P*** [Pa] between the pressure buffer and ambient atmospheric pressure is measured using a differential pressure sensor (model: SDP810-500Pa, manufacturer: Sensirion (Stäfa, Switzerland); measurement range: ±500 Pa, accuracy: 3%, and communication interface: I^2^C). The sign of **Δ*P*** is defined with reference to ambient pressure.

The flow rate inside the tubing connecting the respiratory airflow simulator to the pressure buffer is measured using a flow sensor (model: SFM3000-200-C, manufacturer: Sensirion (Stäfa, Switzerland); measurement range: ±200 L/min, accuracy: 1.5%, response time: 0.5 ms, and communication interface: I^2^C).

The pump in the supply unit is driven via PWM control (duty value: 0–255 and frequency: 35 kHz) using a dedicated controller (model: ESP32, manufacturer: Seeed Studio, Shenzhen, China) and a MOSFET driver circuit (model: EKI04036, manufacturer: Sanken Electric Co., Saitama, Japan).

In this experiment, both the differential pressure sensor and the flow sensor are connected to the controller via I^2^C communication. The controller is connected to a measurement PC via USB (Virtual COM Port). Various parameters—such as timestamp, differential pressure, flow rate, PWM duty value, and the flow rates from the supply unit ***Q*_1_** and the exhaust unit ***Q*_2_** as calculated from the regression equations—are sequentially recorded on the PC according to the controller’s processing cycle (10 ms). The regression equations for calculating ***Q*_1_** and ***Q*_2_** are shown in the next section.

## 3. Modeling of Airflow Characteristics and Its Evaluations

### 3.1. Derivation of Regression Equations for Supply and Exhaust Flow Rates

As a preparatory step, Figure 4 presents the results of the measurements taken to evaluate the dependence of the supply flow rate ***Q*_1_** of the supply unit on the differential pressure **Δ*P*** and PWM duty value **d** under a fixed applied voltage ***V*** = 12.00 V to the amplification circuit of the controller. The measurements were performed using the apparatus and method described in [35]. The relationship between **Δ*P*** and ***Q*_1_** was measured for duty values **d** = 255, 192, 128, 64, 32, and 16. The coefficients of the regression equation ***Q*_1_** (**Δ*P***, **d**) were determined using the least squares method and are given as follows:***Q*_1_** (**Δ*P***, **d**) = −6.2336 × 10^−7^ **Δ*P***^2^ **d**^2^ + 3.3933 × 10^−4^ **Δ*P* d**^2^ − 4.9670 × 10^−2^ **d**^2^+ 9.0562 × 10^−5^ **Δ*P***^2^ **d** − 6.7961 × 10^−2^ **Δ*P* d** + 1.3885 × 10 **d**+ 8.3440 × 10^−3^ **Δ*P***^2^ − 3.7943 × 10^0^ **Δ*P*** + 1.1541 × 10^2^

For appropriate ranges of **Δ*P***, the curves calculated from this regression equation for duty values **d** = 255, 192, 128, 64, 32, and 16 are also plotted in Figure 4.

Regarding the exhaust filter (a thin nonwoven fabric, approximately 29 mm in diameter), prior experiments confirmed that the exhaust flow rate ***Q*_2_** is proportional to the differential pressure **Δ*P***. However, due to significant time-dependent variations, immediately before the present experiment, the respiratory airflow simulator shown in Figure 1 was disconnected. A steady-state condition was established by fixing the applied voltage to ***V*** = 12.00 V and duty value to **d** = 192 for the supply unit. Under the assumption that the supply and exhaust flow rates are equal in this steady state, the average values of supply flow rate ***Q*_1_** and differential pressure **Δ*P*** were measured as 204.3 L/min and 216.5 Pa, respectively. Based on this measurement, the regression equation for the exhaust flow rate ***Q*_2_** (**Δ*P***) was derived as follows:***Q*_2_** (**Δ*P***) = 0.943 **Δ*P*** [L/min]

This measurement point (***Q*_2_**, **Δ*P***) = (***Q*_1_**, **Δ*P***) = (204.3, 216.5), along with the above regression line ***Q*_2_** (**Δ*P***) = 0.943 **Δ*P***, is also shown in Figure 4.

The regression equations ***Q*_1_** (**Δ*P***, **d**) for supply flow and ***Q*_2_** (**Δ*P***) for exhaust flow were both obtained under static conditions—i.e., with constant pump output and constant flow rates throughout the system. In the next section, we attempt to estimate the disturbance flow that simulates respiratory airflow based on these regression equations under the condition where the pump duty value d remains constant. In the following section, we further extend the estimation to the condition where the pump duty value d is actively modulated for differential pressure control.

### 3.2. Estimation of Disturbance Flow Without Supply Pump Control

Using the experimental apparatus shown in Figure 1, the supply unit was operated in a steady state with an applied voltage ***V*** = 12.00 V and duty value **d** = 192. Under these conditions, disturbance flow ***Q*_3_** simulating respiratory airflow was introduced into the pressure buffer using the respiratory airflow simulator. Measurements were taken for 2000 samples over a total duration of 2 s at a sampling interval of 10 ms.

The measured flow rate by the flow meter is denoted as ***Q*_3m_**. The supply and exhaust flow rates, ***Q*_1_** and ***Q*_2_**, were calculated using the previously described regression equations. Flow directions are defined as follows: flow into the pressure buffer (***Q*_1_**, ***Q*_3_**, ***Q*_3m_**) is positive, while flow out of the pressure buffer (***Q*_2_**) is positive. Assuming no leakage from the pressure buffer and negligible flow estimation error, the relationship ***Q*_2_** = ***Q*_1_** + ***Q*_3_** holds. Therefore, the estimated disturbance flow ***Q*_3e_** can be expressed as outlined below:***Q*_3e_** = ***Q*_2_** − ***Q*_1_**

Figure 5 shows the time series of differential pressure **Δ*P***, measured disturbance flow ***Q*_3m_**, PWM duty value d, estimated supply flow ***Q*_1_** (**Δ*P***, **d**), estimated exhaust flow ***Q*_2_** (**Δ*P***), and estimated disturbance flow ***Q*_3e_** (=***Q*_2_** − ***Q*_1_**) derived from the regression equations.

In Figure 5, the four cycles of the disturbance flow ***Q*_3m_** span 13.9 s, from approximately ***t*** = 2.3 s to ***t*** = 16.2 s. This corresponds to a respiratory cycle of 3.5 s. The average half-cycle flow volume (time-integrated, representing either exhalation or inhalation) is 5.6 L. Therefore, the simulator operates at approximately 17.3 breaths per minute with a tidal volume of 1.4 L per breath.

At complete rest, healthy adults typically breathe at approximately 12 breaths per minute with a tidal volume of 0.5 L. During moderate exercise, this increases to approximately 30 breaths per minute with a tidal volume of 3.5 L [30]. Thus, the experimental condition in Figure 5 corresponds to a respiratory airflow level between complete rest and moderate exercise but is closer to complete rest.

From Figure 5, a positive correlation is observed between the measured disturbance flow ***Q*_3m_** and the differential pressure **Δ*P***. That is, when ***Q*_3m_** is positive—representing exhalation—the pressure inside the pressure buffer increases. Conversely, when ***Q*_3m_** is negative—representing inhalation—the pressure decreases. This behavior reflects typical conditions within a PAPR hood.

Furthermore, the time series of measured disturbance flow ***Q*_3m_** and estimated disturbance flow ***Q*_3e_** (=***Q*_2_** (**Δ*P***) − ***Q*_1_** (**Δ*P***, **d**)) in Figure 5 exhibit a good qualitative agreement. This indicates that, under constant pump voltage ***V*** and duty value **d**, simulated expiratory and inspiratory flows can be estimated using the regression equations for ***Q*_1_** (**Δ*P***, **d**) and ***Q*_2_** (**Δ*P***). In other words, expiratory and inspiratory states can be inferred from the observed differential pressure **Δ*P*** and the duty value **d** at a given time.

### 3.3. Estimation of Disturbance Flow Under Active Control of the Supply Pump

In the experimental setup shown in Figure 1, disturbance flow ***Q*_3_** simulating respiratory airflow was introduced into the pressure buffer using the respiratory airflow simulator. Control was applied to the duty value d of the supply unit’s pump in a direction that suppresses pressure fluctuations within the pressure buffer caused by the disturbance flow. Measurements were conducted over 2 s with a sampling interval of 10 ms for a total of 2000 samples.

During the first 4 s after measurement began, the duty value d was fixed at 192. After that point, control of the duty value **d** was initiated to suppress pressure fluctuations in the pressure buffer caused by the disturbance flow. Specifically, to reduce noise in the control signal, a moving average of the most recent 100 data points of the estimated disturbance flow ***Q*_3e_** (calculated as ***Q*_2_** (**Δ*P***) − ***Q*_1_** (**Δ*P***, **d**)) was computed and denoted as ***Q*_3e_ave_**. This ***Q*_3e_ave_** was used to detect peaks corresponding to exhalation and inhalation.

When a peak corresponding to exhalation was detected, the duty value d was decreased step-by-step toward a target value of **d** = 128 (duty cycle 50%) by reducing it by 2% of the difference between the current value dc and the target at each step. Conversely, when a peak corresponding to inhalation was detected, the duty value d was increased step-by-step toward a target value of **d** = 255 (duty cycle 100%) by increasing it by 2% of the difference between the current value **d_c_** and the target at each step. This control represents a simple form of feedforward control, taking into account the response delay between the change in duty value and the corresponding change in differential pressure. The procedure for controlling the duty value d based on ***Q*_3e_ave_** is as follows:(1)The moving average of the past 100 ***Q*_3e_** values, ***Q*_3e_ave_**, is calculated in every step in the time loop;(2)After the positive peak of ***Q*_3e_ave_** is detected, the duty value **d** is reduced by (**d_c_** − 128)/50 at every step, where **d_c_** is the current value of **d** in that step;(3)After the negative peak of ***Q*_3e_ave_** is detected, the duty value **d** is increased by (255 − **d_c_**)/50 at every step, where **d_c_** is the current value of **d** in that step.

Figure 6 shows the time series of differential pressure **Δ*P***, measured disturbance flow ***Q*_3m_**, duty value d, estimated supply flow ***Q*_1_** (**Δ*P***, **d**), estimated exhaust flow ***Q*_2_** (**Δ*P***), estimated disturbance flow ***Q*_3e_** (=***Q*_2_** − ***Q*_1_**), and the moving average of ***Q*_3e_** over the last 100 samples, denoted as ***Q*_3e_ave_**.

In Figure 6, the three cycles of disturbance flow ***Q*_3m_** from approximately ***t*** = 6.4 s to ***t*** = 16.6 s span is about 10.3 s. The half-cycle flow volume (corresponding to either exhalation or inhalation) is 4.2 L. Therefore, the simulator exchanges approximately 23.4 breaths per minute with a tidal volume of 1.4 L per breath. The experimental condition can be regarded as simulating respiratory airflow at a level of physical activity between complete rest and moderate exercise but is closer to complete rest [30].

Before the control was applied (***t*** < 4 s), a positive correlation was observed between the measured disturbance flow ***Q*_3m_** and the differential pressure **Δ*P***. That is, when ***Q*_3m_** is positive (representing exhalation), the pressure inside the pressure buffer increases, and when ***Q*_3m_** is negative (representing inhalation), the pressure decreases. This behavior reflects the typical internal environment of a PAPR hood.

At the time approximately one second after the control was applied (***t*** > 5 s) from the time it was applied (***t*** = 4 s), the fluctuation amplitude of **Δ*P*** was clearly reduced. This is considered to be the result of controlling the duty value d based on ***Q*_3e_ave_**, as described above.

In Figure 6, the time series of measured disturbance flow ***Q*_3m_** and estimated disturbance flow ***Q*_3e_** (=***Q*_2_** − ***Q*_1_**) show good qualitative agreement regardless of whether duty control based on ***Q*_3e_ave_** was applied. This indicates that even when the duty value d is being actively modulated, the simulated expiratory and inspiratory flows can still be reasonably estimated using the regression-based flow models ***Q*_1_** (**Δ*P***, **d**) and ***Q*_2_** (**Δ*P***). In other words, the expiratory and inspiratory states can be inferred from the differential pressure ΔP and the duty value d at each time point.

## 4. Discussion

### 4.1. Validation of Regression Models Under Dynamic Conditions

The regression equations for the supply flow rate ***Q*_1_** (**Δ*P***, **d**) and the exhaust flow rate ***Q*_2_** (**Δ*P***), developed in this study, were both derived under static conditions in which the pump output and local flow rates were constant. The fact that these regression models were also found to be valid under dynamic conditions—such as those shown in Figure 5 and Figure 6—is noteworthy.

However, even if the dynamic effects on the differential-pressure-dependent behavior of the exhaust filter flow ***Q*_2_** (**Δ*P***) are expected to be small, quantitative evaluation remains necessary. Furthermore, for the supply unit flow ***Q*_1_** (**Δ*P***, ***V***, **d**)—which depends on the differential pressure, the applied voltage ***V***, and the PWM duty value **d**—it is anticipated that time lags will occur before the fan (pump) speed and flow respond to changes in ***V*** and **d**. Therefore, evaluating the dynamic characteristics of the exhaust filter and supply unit, and incorporating them into the model, is essential for accurately reproducing and predicting the overall system behavior.

### 4.2. Incorporation and Extension of Dynamic Characteristics into the Flow Rate Models

To faithfully reproduce and predict system behavior, it is necessary to evaluate and incorporate the dynamic characteristics of the exhaust filter flow ***Q*_2_** (**Δ*P***) and the supply unit flow ***Q*_1_** (**Δ*P***, ***V***, **d**) into the modeling framework.

When modeling the dynamic behavior of ***Q*_2_** (**Δ*P***), one possible first-order approximation is to introduce the time derivative (d/d***t***)(**Δ*P***) and the integral ∫**Δ*P*** d***t*** of the differential pressure. The time derivative (d/d***t***)(**Δ*P***) captures the system’s responsiveness to rapid pressure changes, allowing for consideration of elastic responses of the filter’s fiber structure and inertial effects of air. The integral ∫**Δ*P*** d***t*** accounts for history-dependent effects such as gradual deformation of fibers (creep) or structural changes due to long-term exposure to pressure, enabling the modeling of slowly evolving flow resistance or clogging.

Similarly, when extending the model of ***Q*_1_** (**Δ*P***, ***V***, **d**) to include dynamic characteristics, the following time-dependent terms should be considered: (d/d***t***)(**Δ*P***) and ∫**Δ*P*** d***t*** for pressure dynamics; (d/d***t***)(***V***) and ∫***V*** d***t*** for voltage dynamics; and (d/d***t***)(**d**) and ∫**d** d***t*** for PWM control dynamics.

Specifically, (d/d***t***)(**Δ*P***) reflects inertial or acceleration effects caused by changes in pressure, while ∫**Δ*P*** d***t*** reflects delayed elastic responses. The term (d/d***t***)(***V***) reflects response delays caused by acceleration or deceleration of the fan, and ∫***V*** d***t*** represents accumulated effects analogous to capacitive behavior. The term (d/d***t***)(**d**) captures fan response delays to sudden PWM signal changes, while ∫**d** d***t*** reflects accumulated control history, indirectly representing fan speed memory.

By linearly combining these dynamic terms with the original regression equations obtained under static conditions, a new regression model that incorporates dynamic characteristics can be constructed. Such an extended model is expected to more accurately describe the system’s behavior in response to respiratory flow.

### 4.3. Preliminary Evaluation of the Control Algorithm and Future Perspectives

Figure 7 shows the relationship between the measured disturbance flow ***Q*_3m_** and the differential pressure **Δ*P*** before (***t*** < 4 s), and after one second has passed since the control was turned on (**t** > 5 s), for the respiratory airflow generated manually using the respiratory airflow simulator shown in Figure 6. Prior to control (***t*** < 4 s), the R-squared value of the regression line was 0.86, indicating a strong positive correlation between the disturbance flow and the pressure variation. This behavior is consistent with that seen in typical face masks and PAPRs, in which pressure resistance occurs during breathing: pressure in the buffer increases during exhalation and decreases during inhalation. In contrast, after control was applied (***t*** > 5 s), the R-squared value dropped to 0.22, showing a weak or negligible correlation. Regarding the fluctuation in the differential pressure, before the control was applied (t < 4 s), the mean value was 203.5 [Pa] and the standard deviation was 9.0 [Pa], whereas after the control was applied (t > 5 s), the mean value was 197.0 [Pa] and the standard deviation was 4.9 [Pa], showing a reduction of approximately 45%. This suggests that the introduced control algorithm effectively suppressed the generation of pressure resistance during breathing.

The control algorithm implemented in this study was designed to evaluate how well the static regression models for ***Q*_1_** and ***Q*_2_** can follow changes in PWM duty value in response to respiratory airflow fluctuations. Thus, the performance of pressure control itself is beyond the primary scope of this study.

In the future, it will be important to improve the accuracy of the flow models and incorporate dynamic characteristics while also advancing pressure control algorithms aimed at supporting breathing. If the pressure control that assists breathing—namely, depressurizing upon exhalation and pressurizing upon inhalation—can be realized, it may become possible to develop a PAPR that actually makes breathing easier for the wearer.

For more effective control of respiratory assist pressure, it is desirable to install pumps not only on the supply side but also on the exhaust side to enable differential pressure control. With such a configuration, the internal pressure of the PAPR hood can be actively controlled relative to the external environment, both in the positive- and negative-pressure directions. In systems where the interior of the PAPR is maintained at positive pressure at all times, a control strategy that sets a slight positive pressure (e.g., approximately 5 Pa) upon detection of an expiratory state and a higher positive pressure (e.g., approximately 200 Pa) upon detection of an inspiratory state is suitable. On the other hand, if the gap between the sealing part of the PAPR hood and the face can be made sufficiently small, and if the inflow of outside air through this gap under negative pressure can be tolerated, a control strategy that applies negative pressure during exhalation and positive pressure during inhalation can also be employed.

Since respiratory motion generally exhibits features such as a time-averaged airflow close to zero between exhalation and inhalation, periodic alternation between inhalation and exhalation, and a rough correlation of tidal volume and respiratory cycle with physical activity levels [32], predictive control based on these characteristics is considered to be effective in many cases.

As shown in Figure 5 and Figure 6, the disturbance flow manually generated using the respiratory airflow simulator corresponded to a level of physical activity between complete rest and moderate exercise but is closer to complete rest [32] in terms of respiratory rate and tidal volume. However, in terms of temporal variation in flow, the simulated pattern does not closely resemble actual respiratory patterns [36]. A future challenge will be to modify the respiratory airflow simulator to use a linear motor to more accurately reproduce time-series data of actual respiratory flow. The introduction of an automatic artificial lung device could also be considered.

For public-oriented PAPRs intended as an alternative to lockdown measures, we will continue research and development aimed at improving their wearing comfort.

## 5. Conclusions

In this study, a method was proposed and experimentally validated for estimating respiratory states based on differential pressure **Δ*P*** and PWM control signals (duty value), with the aim of improving the wearing comfort of public-oriented Powered Air-Purifying Respirators (PAPRs) that have drawn increasing attention as countermeasures against airborne infectious diseases.

First, the fluid characteristics of the supply unit and exhaust filter were modeled under static conditions, and regression equations for supply and exhaust flow rates were derived based on **Δ*P*** and the PWM duty value.

Next, using an experimental setup in which simulated respiratory flow was introduced as a disturbance, it was shown that the proposed model can accurately estimate expiratory and inspiratory states. Furthermore, it was demonstrated that by controlling the internal pressure of the PAPR hood based on the estimated respiratory state, the pressure resistance against breathing could be reduced.

A key outcome of this study was the experimental confirmation that, even under dynamic conditions, the behavior of disturbance flow could be qualitatively reproduced using the regression equations developed under static conditions, and that pressure fluctuations were attenuated by applying pressure control for respiratory assistance. These findings demonstrate the potential for realizing a “PAPR that makes breathing easier when worn” through a control scheme that depressurizes during exhalation and pressurizes during inhalation.

Future work includes improving the accuracy of the flow models, incorporating dynamic characteristics, and enhancing the disturbance generation system to reproduce more realistic respiratory flow based on actual respiratory data. These developments are expected to enable evaluation under more practical conditions and contribute to establishing the technical foundation for a comfortable and high-performance PAPR that can be used routinely by the general public.

## Figures and Tables

**Figure 1 sensors-25-02939-f001:**
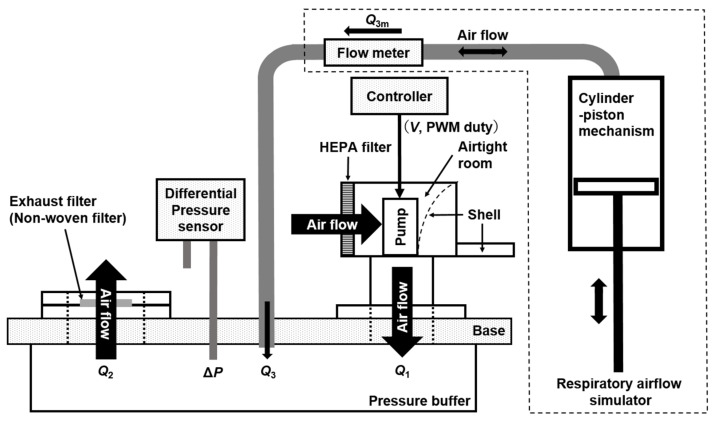
Schematic of the experimental setup used to evaluate the airflow characteristics of the PAPR prototype. The system consists of a supply unit (including a HEPA filter and electric pump), an exhaust unit with a nonwoven fabric filter, a pressure buffer simulating the internal environment of a pressurized PAPR hood, and a respiratory airflow simulator (piston–cylinder mechanism) that mimics human breathing. The supply and exhaust units are connected to the pressure buffer, and airflow parameters are measured to evaluate whether the disturbance flow ***Q*_3_** can be reasonably estimated from the supply flow rate ***Q*_1_** and exhaust flow rate ***Q*_2_**, as calculated by regression models under various conditions.

**Figure 2 sensors-25-02939-f002:**
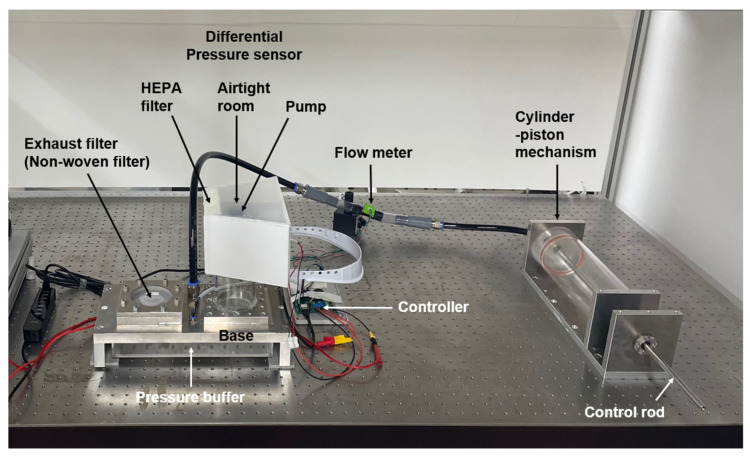
Photograph of the experimental setup used for airflow characterization in the PAPR prototype. The image shows the physical layout of the experimental system, including the supply unit with HEPA filter and electric pump, the exhaust unit with nonwoven fabric filter, the pressure buffer simulating the internal environment of a pressurized PAPR hood, and the respiratory airflow simulator. The components are assembled to allow for flow rate and differential pressure measurements under various conditions.

**Figure 3 sensors-25-02939-f003:**
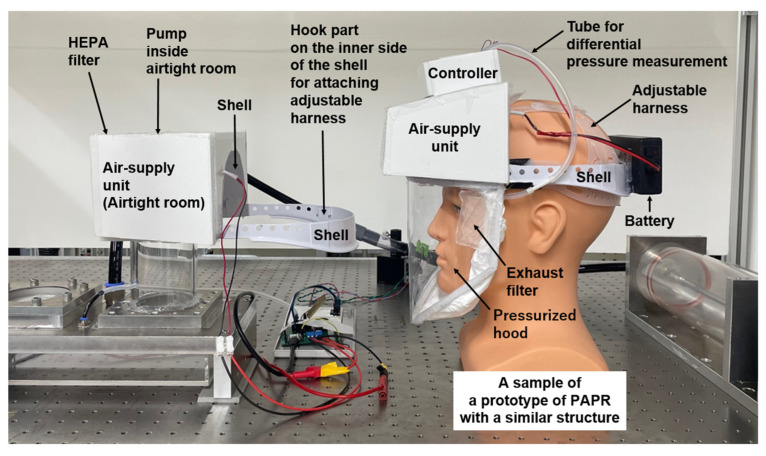
Photograph of a completed PAPR prototype with a configuration similar to the planned final design, placed within the experimental setup. This prototype uses supply and exhaust units similar to those developed and tested in this study and features an overall structural configuration that closely resembles—but is not identical to—the intended final version of the PAPR. It was placed inside the experimental apparatus for illustrative purposes to show how the supply unit and exhaust filter would be mounted in an actual PAPR.

**Figure 4 sensors-25-02939-f004:**
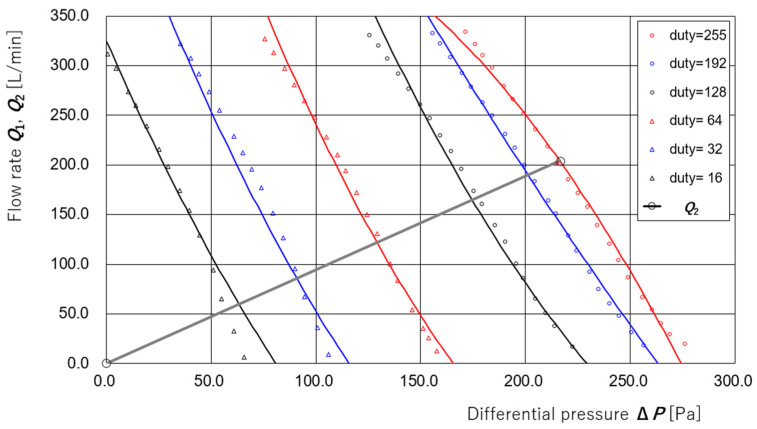
Measurement results and regression curves of the supply flow rate *Q*_1_ as a function of differential pressure **Δ*P*** and PWM duty value **d**. The supply flow rate ***Q*_1_** of the supply unit was measured at an applied voltage ***V*** = 12.00 V for six different PWM duty values (**d** = 255, 192, 128, 64, 32, 16). The data points were used to derive a multivariate regression equation ***Q*_1_** (**Δ*P***, **d**) using the least squares method. The resulting regression curves are overlaid for each duty value. A large gray-color open circle indicates the measurement point used to derive the regression line for the exhaust flow rate ***Q*_2_** (**Δ*P***) = 0.943**Δ*P***, assuming steady-state conditions where ***Q*_1_** ≈ ***Q*_2_**.

**Figure 5 sensors-25-02939-f005:**
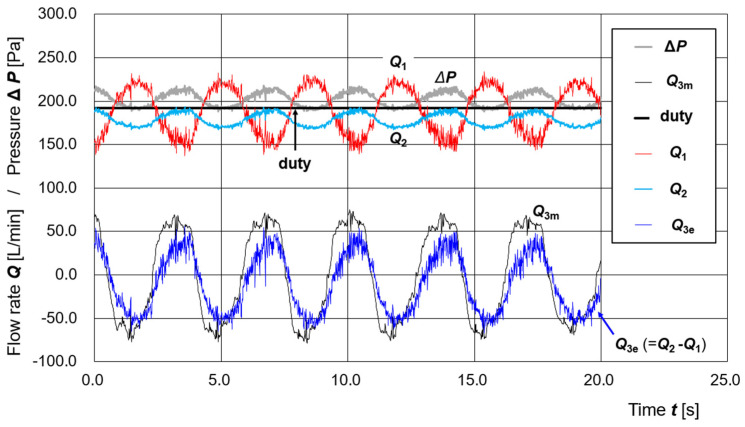
Time series of measured and estimated parameters during simulated respiratory airflow without supply pump control. Under a constant applied voltage ***V*** = 12.00 V and PWM duty value **d** = 192, disturbance flow mimicking respiratory activity was introduced into the pressure buffer using the respiratory airflow simulator. The figure shows the time series of differential pressure **Δ*P***, measured disturbance flow ***Q*_3m_**, PWM duty value **d**, estimated supply flow rate ***Q*_1_** (**Δ*P***, **d**), estimated exhaust flow rate ***Q*_2_** (**Δ*P***), and estimated disturbance flow ***Q*_3e_** (=***Q*_2_** − ***Q*_1_**). The good agreement between ***Q*_3m_** and ***Q*_3e_** demonstrates that expiratory and inspiratory states can be inferred from **Δ*P*** and **d** even under steady-state pump operation.

**Figure 6 sensors-25-02939-f006:**
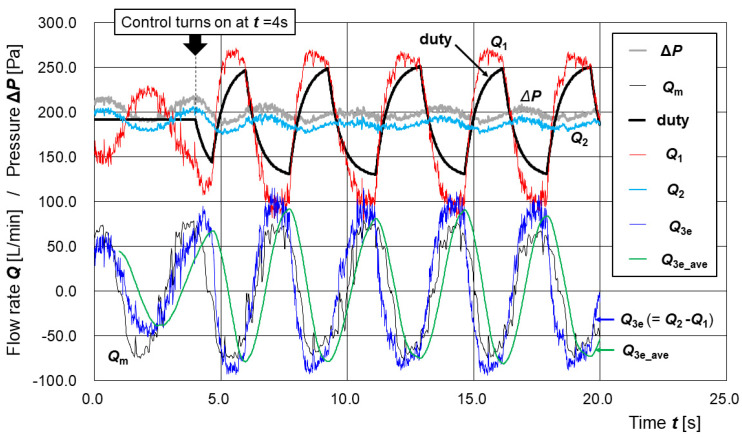
Time series of measured and estimated parameters during simulated respiratory airflow with active control of the supply pump. Disturbance flow mimicking respiratory activity was introduced into the pressure buffer using the respiratory airflow simulator. The supply unit’s pump was actively controlled based on the moving average of the estimated disturbance flow ***Q*_3e_** (***Q*_3e_ave_**), calculated from the regression models of ***Q*_1_** (**Δ*P***, **d**) and ***Q*_2_** (**Δ*P***). The time series of differential pressure **Δ*P***, measured disturbance flow ***Q*_3m_**, PWM duty value d, estimated supply flow ***Q*_1_** (**Δ*P***, **d**), estimated exhaust flow ***Q*_2_** (**Δ*P***), estimated disturbance flow ***Q*_3e_** (=***Q*_2_** − ***Q*_1_**), and ***Q*_3e_ave_** are shown. The results demonstrate that respiratory states can be estimated even when the duty value d is dynamically modulated.

**Figure 7 sensors-25-02939-f007:**
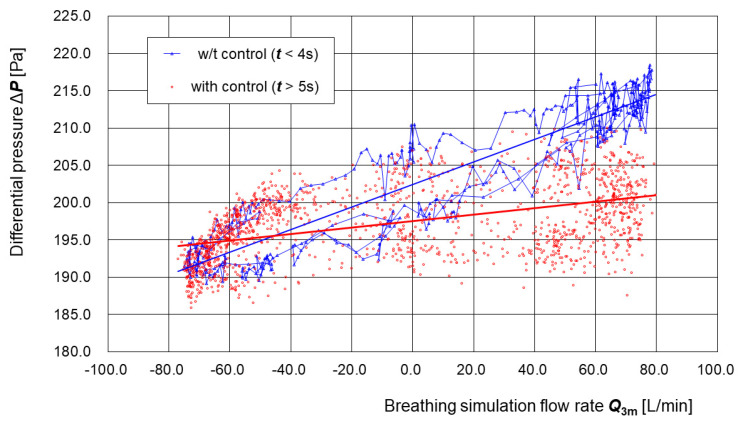
Relationship between measured disturbance flow *Q*_3m_ and differential pressure Δ*P* before and after pressure control. The scatter plot shows the correlation between the differential pressure **Δ*P*** and the measured disturbance flow ***Q_3m_*** before control (***t*** < 4 s) and after control (***t*** > 5 s) using the respiratory airflow simulator. Before control, a strong positive correlation was observed (R-squared = 0.86), indicating that respiratory activity directly affected pressure fluctuations in the buffer. After control was applied, the correlation decreased significantly (R-squared = 0.22), suggesting that the introduced control algorithm successfully reduced pressure resistance during breathing.

**Table 1 sensors-25-02939-t001:** Comparison of specifications between a commercial PAPR (3M) and the low-cost prototype developed in this study.

Model	[A] Versaflo TR-300+ (3M)	[B] A Prototype PAPR
Photo	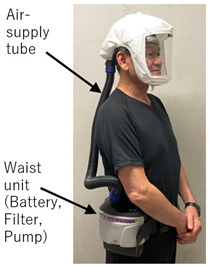	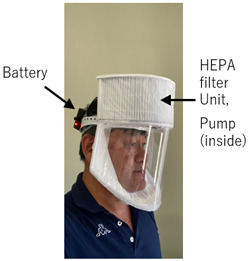
Filter	Nonwoven filter	HEPA filter
Internal Pressure	positive	positive
Flow rate [L/min]	180 (Low-mode) or 200 (High-mode)	400
Cost	Price: USD 1000	Parts cost: USD 40

The commercial PAPR from 3M [24] and the prototype PAPR developed by the authors [25] are compared in terms of key specifications such as filter type, internal pressure, flow rate, and cost. While the 3M PAPR is designed for medical professionals in high-risk environments, the prototype aims to offer comparable performance at significantly lower cost for public use during airborne infectious disease outbreaks.

## Data Availability

The datasets used and/or analyzed in the current study are available from the author on reasonable request.

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
