# Peer review of "Estimation of Respiratory States Based on a Measurement Model of Airflow Characteristics in Powered Air-Purifying Respirators Using Differential Pressure and Pulse Width Modulation Control Signals—In the Development of a Public-Oriented Powered Air-Purifying Respirator as an Alternative to Lockdown Measures"

_sensors, 2025, doi:10.3390/s25092939_

Round 1
Reviewer 1 Report
Comments and Suggestions for Authors
The paper is interesting. I would suggest a minor revision for the following points which are not clear to me:
1) Refs. 21 and 22 is misplaced and should be interchanged.
2) Did you consider vapour content during exhalation and preventing its accumulation inside the mask?
3) Line 365: why "100 data points"? How sensitive is this number?
4) The fluctuations of Delta (P) in Figs. 5 & 6 should be compared quantitatively.
5) It is still not clear how Delta(p)affects the exhalation and inhalation? Would a higher Delta(p) or lower Delta(p) be better for easing the breathing?
Author Response
Reply to Comments by Reviewer#1
We sincerely thank the reviewer for the thoughtful and constructive feedback, which has helped improve the quality of the manuscript. Please find our detailed responses below:
The paper is interesting. I would suggest a minor revision for the following points which are not clear to me:
1) Refs. 21 and 22 is misplaced and should be interchanged.
→ The references have been corrected as suggested.
2) Did you consider vapour content during exhalation and preventing its accumulation inside the mask?
→ Thank you for this important point. Lines L139–L147 have been revised as follows to clarify our consideration of vapor and COâ‚‚ accumulation:
The flow rates of the PAPRs shown in Table 1 range from 180 to 200 [L/min] for com-mercially available units, and 400 [L/min] for the prototype developed by the au-thors—values significantly higher than the resting respiratory flow rate of an adult male (ap-proximately 6 L/min) [30]. This high flow rate is intended to prevent the accumulation of exhaled carbon dioxide within the PAPR hood. With such high air supply rates, these PAPRs achieve relatively comfortable wearing conditions without fogging of the face shield due to exhaled moisture. However, such high flow rates can also be considered a substantial waste, suggesting that flow path design is critical to ensure the rapid expulsion of exhaled air from the hood.
3) Line 365: why "100 data points"? How sensitive is this number?
→ The moving average window size of 100 was selected arbitrarily. Since we used a peak detection method that references one point before and after each candidate peak, a moderately large window was chosen for stability. In our tests, varying the window size among 50, 100, and 150 resulted in no substantial difference.
4) The fluctuations of Delta (P) in Figs. 5 & 6 should be compared quantitatively.
→ A quantitative comparison of ΔP fluctuations before and after control is presented in Figure 7, which is based on the data shown in Figure 6. Please refer to Lines L471–L485 as shown below:
Figure 7 shows the relationship between the measured disturbance flow Q3m and the differential pressure ΔP before (t < 4 s), and after one second has passed since the control was turned ON (t > 5 s), for the respiratory airflow generated manually using the Respira-tory Airflow Simulator shown in Figure 6. Prior to control (t < 4 s), the R-squared value of the regression line was 0.86, indicating a strong positive correlation between the disturb-ance flow and the pressure variation. This behavior is consistent with that seen in typical face masks and PAPRs, in which pressure resistance occurs during breathing: pressure in the buffer increases during exhalation and decreases during inhalation. In contrast, after control was applied (t > 5 s), the R-squared value dropped to 0.22, showing a weak or neg-ligible correlation. Regarding the fluctuation of the differential pressure, before the control was applied (t < 4 s), the mean value was 203.5 [Pa] and the standard deviation was 9.0 [Pa], whereas after the control was applied (t > 5 s), the mean value was 197.0 [Pa] and the standard deviation was 4.9 [Pa], showing a reduction of approximately 45%. This sug-gests that the introduced control algorithm effectively suppressed the generation of pres-sure resistance during breathing.
5) It is still not clear how Delta(p)affects the exhalation and inhalation? Would a higher Delta(p) or lower Delta(p) be better for easing the breathing?
=> During the exhalation phase, a low pressure is desirable to assist the act of exhaling. During the inhalation phase, a high pressure is desirable to assist the act of inhaling. In the present setup, since the pump is installed only on the supply side, only pressurization is possible. Therefore, the best achievable operation is as follows:
â‘ During the exhalation phase, the pump output is reduced to create a slight positive pressure.
â‘¡ During the inhalation phase, the pump output is increased to achieve as high a positive pressure as possible.
To facilitate reader understanding, the second-to-last paragraph of the abstract was revised as follows.
Furthermore, based on the estimated respiratory states, a theoretical investigation was conducted on constant pressure control inside the PAPR and on inverse pressure control aimed at supporting respiratory activity—namely, pressure control that assists breathing by depressurizing when expiratory motion is detected and pressurizing when inspiratory motion is detected.
Reviewer 2 Report
Comments and Suggestions for Authors
This article addresses the limitations of lockdown measures in the prevention and control of airborne diseases and proposes to promote the popularization of PAPRs by optimizing their wearing comfort, which has important practical significance. The manuscript demonstrates clear structure, coherent organization and comprehensive content. However, the following revisions are necessary before it can be considered for acceptance.
- The introduction could more explicitly discuss the limitations of existing PAPR techniques and how the proposed method overcomes these limitations, which would better highlight the novelty of this study.
- In this paper, it only provided a brief review of PAPR-related research but lacked a detailed discussion of the latest research progress directly related to air filtration. It is recommended to incorporate some relevant references, such as Atmos. Pollut. Res., 2023, 14, 101840, Adv. Mater., 2020, 32, 2002361, Membranes, 2021, 11, 250.
- Some figures (e.g., Figures 5 and 6) are dense and could benefit from more precise labeling and elaborative captions to enhance their readability and interpretability.
- This manuscript presented experimental data and analyzed the performance of the regression equations under simulated respiratory airflow. It is suggested to increase the statistical analysis of the experimental data to enhance the reliability and persuasiveness of the results.
- It should be emphasized that a comparison with existing commercial PAPR technologies would further emphasize the novelty of this study.
- The format of the references is inconsistent. Some references, such as Reference 2 and 10, do not display the DOI in color, whereas others are presented in blue. Please check the format of the references carefully to ensure consistency.
Author Response
Reply to Comments by Reviewer#2
We sincerely thank the reviewer for the thoughtful and constructive feedback, which has helped improve the quality of the manuscript. Please find our detailed responses below:
This article addresses the limitations of lockdown measures in the prevention and control of airborne diseases and proposes to promote the popularization of PAPRs by optimizing their wearing comfort, which has important practical significance. The manuscript demonstrates clear structure, coherent organization and comprehensive content. However, the following revisions are necessary before it can be considered for acceptance.
The introduction could more explicitly discuss the limitations of existing PAPR techniques and how the proposed method overcomes these limitations, which would better highlight the novelty of this study.
→The final block of the Introduction has been revised as follows.
In this study, we experimentally derive a fluid dynamic model—i.e. a calibration formula that expresses the flow rate Q of the supply and exhaust units as a function of pump duty and differential pressure ΔP—with the aim of enabling real-time pressure control synchronized with respiratory activity. This pressure control assists breathing by shifting the internal pressure inside the hood to a lower level during exhalation and to a higher level during inhalation, based on the detection of respiratory phases. Furthermore, we attempt to detect respiratory activity (i.e., exhalation and inhalation) based on the de-rived model.
In this paper, it only provided a brief review of PAPR-related research but lacked a detailed discussion of the latest research progress directly related to air filtration. It is recommended to incorporate some relevant references, such as Atmos. Pollut. Res., 2023, 14, 101840, Adv. Mater., 2020, 32, 2002361, Membranes, 2021, 11, 250.
→ The suggested references are added.
- Kumar, S. Dhawan, M. V. Kumar, M. Khare, S. M. S. Nagendra, S. K. Dubey, and D. S. Mehta, “Detection and identification of shape, size, and concentration of particulate matter in ambient air using bright field microscopy-based system,” Atmospheric Pollution Research, vol. 14, no. 11, p. 101913, 2023. https://doi.org/10.1016/j.apr.2023.101913
- Zhang, H. Liu, N. Tang, S. Zhou, J. Yu, and B. Ding, “Spider-web-inspired PM0.3 filters based on self-sustained electrostatic nanostructured networks,” Advanced Materials, vol. 32, no. 29, p. 2002361, Jul. 2020.
- https://advanced.onlinelibrary.wiley.com/doi/full/10.1002/adma.202002361
- K. Essa, S. A. Yasin, I. A. Saeed, and G. A. M. Ali, “Nanofiber-based face masks and respirators as COVID-19 protection: A review,” Membranes, vol. 11, no. 4, p. 250, Mar. 2021.https://www.mdpi.com/2077-0375/11/4/250
Some figures (e.g., Figures 5 and 6) are dense and could benefit from more precise labeling and elaborative captions to enhance their readability and interpretability.
→The vertical and horizontal axes in Figure 7 have been reversed to facilitate understanding of the phenomenon.
The description related to Figure 7 has been revised as follows (L476–490).
Figure 7 shows the relationship between the measured disturbance flow Q3m and the differential pressure ΔP before (t < 4 s), and after one second has passed since the control was turned ON (t > 5 s), for the respiratory airflow generated manually using the Respira-tory Airflow Simulator shown in Figure 6. Prior to control (t < 4 s), the R-squared value of the regression line was 0.86, indicating a strong positive correlation between the disturb-ance flow and the pressure variation. This behavior is consistent with that seen in typical face masks and PAPRs, in which pressure resistance occurs during breathing: pressure in the buffer increases during exhalation and decreases during inhalation. In contrast, after control was applied (t > 5 s), the R-squared value dropped to 0.22, showing a weak or neg-ligible correlation. Regarding the fluctuation of the differential pressure, before the control was applied (t < 4 s), the mean value was 203.5 [Pa] and the standard deviation was 9.0 [Pa], whereas after the control was applied (t > 5 s), the mean value was 197.0 [Pa] and the standard deviation was 4.9 [Pa], showing a reduction of approximately 45%. This sug-gests that the introduced control algorithm effectively suppressed the generation of pres-sure resistance during breathing.
This manuscript presented experimental data and analyzed the performance of the regression equations under simulated respiratory airflow. It is suggested to increase the statistical analysis of the experimental data to enhance the reliability and persuasiveness of the results.
→ In (L476–490), the mean and standard deviation of the differential pressure before and after the start of control were presented, and the reasons for these values were discussed.
It should be emphasized that a comparison with existing commercial PAPR technologies would further emphasize the novelty of this study.
→ Regarding the comparison, especially in terms of cost, the text has been revised as follows.
The prototype PAPR [25] developed in this study incorporates a HEPA filter (cut from a commercially available air purifier filter), a pump, and a battery, with a total component procurement cost of approximately USD 40. The prototype was assembled at Cebu Tech-nological University (Cebu city, Philippines), and the components were procured through online shopping sites (such as Lazada). As a result, the cost of components was kept low. The device supplies filtered air, purified via a HEPA filter capable of removing over 99.97% of 0.3 μm particles, into the hood to maintain a positive pressure environment. Consequently, even if small gaps exist between the hood and the user’s face, outside air inflow can be physically prevented. Exhaust is released passively through the exhaust fil-ter via differential pressure, which may offer some virus emission suppression if the wearer is infected. This design achieves the core functionality of a commercial medi-cal-grade PAPR.
Commercial medical PAPRs (e.g., 3M [24], CleanSpace [27,28]) generally consist of nonwoven fabric filters, pumps, and batteries in a relatively simple structure, yet they are often priced above USD 1,000. This high cost is likely due to their intended use in high-risk healthcare environments and the lack of mass production. During the COVID-19 pan-demic, low-cost fabrication of PAPR-equivalent devices using 3D printing has also been proposed [29].
The format of the references is inconsistent. Some references, such as Reference 2 and 10, do not display the DOI in color, whereas others are presented in blue. Please check the format of the references carefully to ensure consistency.
→ Format of references has been corrected.

Reviewer 3 Report
Comments and Suggestions for Authors
Reducing and preventing the spread of pathogens is very important, especially in the context of the recent COVID19 pandemic. Scientists from many countries have developed many interesting methods to minimize the risk of infection. Extensive mathematical models have been created to determine the numerical value of the probability of infection (exposure to a given emission level expressed using the quanta number). It is very important and obvious that the most effective actions are or should be directed at controlling the sources of pathogen emission, and not, as is the case in many cases, only at the preparation of contaminated air.
The article fits into the extensive group of studies on preventive measures.
The structure of the article is far from the standard one. Paragraphs regarding the measurement method were placed in the introduction.
The topic is ambitious, necessary and timely, but the content of the manuscript does not allow understanding the meaning of the measurements conducted.
In my opinion, a number of substantive errors that discredit the significance of the obtained results.
I also have also a few additional particular comments to the content of the presented article. The order of the comments does not reflect their significance. It results only from the order of appearance in the text of the article.
My remarks and comments:
1. Lines 45-46, “depressurization during expiration and pressurization during inspiration” - in the literature, the breathing cycle is named exhalation and inhalation, and optionally break; I suggest keeping this nomenclature
2. Line 58, “fluid dynamic model” - is it CFD model or another one?
3. Line 74, “is difficult to control” – it is really hard to agree with this statement; the literature describes methods of controlling the sources of pathogens, and even interesting devices and equipment for medical awards have been developed; devices that intercept pathogens immediately after their emission and prevent their spread. I recommend more in-depth literature studies.
4. Line 102, “NIOSH” - although this shortcut is quite obvious, but for clarity, it should be explained.
5. Line 105, “APF of 1,000” - it must be given for which fraction, for which particle size. this is very important information.
6. Table 1 - the flow rate [L/min] line contains values that are many times higher than the human breathing flow rate; studies typically assume air consumption of 6L/min; the maximum flow rate I have seen was 18L/min for an athlete with very high effort. Shouldn't it be L/h?
7. Line 128, “mass production” - Was the declared $40 calculated for mass production?
8. Line 133, “droplet” - for accuracy, the diameter range of pathogenic particles for which this assumption applies should be specified.
9. Line 136, “with Rt_target = 0.8” - It would be very interesting to know what infection risk model was used to determine the reproduction rate.
10. Line 136, “the waring rate is 100%” - requires clarification of what this term means; does it mean that the entire population will use PAPR.
11. Lines 156-158, “The air supply flow is changed by adjusting the speed of the variable-frequency centrifugal fan according to the air velocity at the half mask outlet monitored by the modular wind speed transmitter.” - this sentence contradicts itself. According to this sentence - The fan speed was controlled by air flow and simultaneously by air speed (not velocity, it is not the same). After all, it is the same, assuming that the physicochemical parameters of the air did not change significantly.
12. Line 161, “In the future” - Most systems currently in use already use this solution.
13. Line 173, “mimics human breathing” - what the piston displacement was, whether it was a uniform movement (rectangular course), or a variable sinusoidal or, real based on previously collected averaged measurements of the flow rate of a real person. This is of great importance for leak simulation, as shown by the results of research in already published articles. A sudden change in flow, causing a sudden change in the relative pressure under the mask, causes non-linear, inertial changes in the effective cross-section of the gaps, the degree of mask fit to the face.
14. Lines 203-204, “mimics the respiratory flow” - What was the flow rate, what was the length of each phase of the breathing cycle, was there a break between exhalation and inhalation? The standard cycle is 6L/min, 2.5s inhalation, 2.5s of exhalation and 1s break.
15. Line 237, “is driven manually” - I strongly encourage you to improve this method; why wasn't an artificial lung system used, as in most similar studies?
16. Line 248, “EPS32” - Is this really EPS32 or isn't it the ESP32 popular among young people?
17. Line 254, “calculated from the regression equations” - requires an explanation of exactly what equations are meant.
18. Lines 259-260, “the dependence of the supply flow rate Q1 of the supply unit on the differential pressure ΔP and PWM duty value d” - so far it could be understood that the air intake fan was to be controlled by changing the PWM duty cycle?
19. Figure 4 - It's hard to understand how it's possible that for zero flow there's a pressure difference?
20. Figure 5 - did Q1 and Q2 take on a value of about 200L/min? that is really a lot of air. How huge would the pressure drop have to be on the filter material, even with such a large surface area as shown earlier in Figure 3.
Author Response
Reply to Comments by Reviewer#3
We sincerely thank the reviewer for the thoughtful and constructive feedback, which has helped improve the quality of the manuscript. Please find our detailed responses below:
Reducing and preventing the spread of pathogens is very important, especially in the context of the recent COVID19 pandemic. Scientists from many countries have developed many interesting methods to minimize the risk of infection. Extensive mathematical models have been created to determine the numerical value of the probability of infection (exposure to a given emission level expressed using the quanta number). It is very important and obvious that the most effective actions are or should be directed at controlling the sources of pathogen emission, and not, as is the case in many cases, only at the preparation of contaminated air.
The article fits into the extensive group of studies on preventive measures.
The structure of the article is far from the standard one. Paragraphs regarding the measurement method were placed in the introduction.
The topic is ambitious, necessary and timely, but the content of the manuscript does not allow understanding the meaning of the measurements conducted.
In my opinion, a number of substantive errors that discredit the significance of the obtained results.
I also have also a few additional particular comments to the content of the presented article. The order of the comments does not reflect their significance. It results only from the order of appearance in the text of the article.
→To ensure that the significance of the measurement is understood and to prevent any misunderstandings, we have provided an explanation in response to the following comment and revised the main text accordingly.
My remarks and comments:
- Lines 45-46, “depressurization during expiration and pressurization during inspiration” - in the literature, the breathing cycle is named exhalation and inhalation, and optionally break; I suggest keeping this nomenclature
→As suggested, we have standardized the terminology to "exhalation" and "inhalation." Additionally, we have added the following explanation of terms in L86–89.
In general, there is a trade-off between aerosol filtration efficiency and airflow resistance in nonwoven filters; however, new types of filters based on novel principles that overcome this limitation have also been developed [18,19].
- Line 58, “fluid dynamic model” - is it CFD model or another one?
→The model is a fluid dynamic model as described in “3. Modeling of Airflow Characteristics and its Evaluations”.
- Line 74, “is difficult to control” – it is really hard to agree with this statement; the literature describes methods of controlling the sources of pathogens, and even interesting devices and equipment for medical awards have been developed; devices that intercept pathogens immediately after their emission and prevent their spread. I recommend more in-depth literature studies.
→ The description in L77–82 has been revised to a more modest expression as follows.
Airborne transmission, alongside contact, oral, and droplet transmission, is a major route of infection. Comparing the other transmission routes, airborne transmission is more difficult to control due to the wide dispersion of aerosols in the air [11,12], and is considered a likely primary route in the next pandemic. At present, apart from vaccination, infection control options available to the general public are limited and often insufficient [13,14].
- Line 102, “NIOSH” - although this shortcut is quite obvious, but for clarity, it should be explained.
→ At L110-11, The full name has been added as follows:
by NIOSH (National Institute for Occupational Safety and Health)
- Line 105, “APF of 1,000” - it must be given for which fraction, for which particle size. this is very important information.
→ This is a very important point. It appears that the evaluation of filtration efficiency is generally based on the MPPS of 0.3 micrometers; however, some references do not clearly specify the particle size. Therefore, although some ambiguity remains, we have revised the explanation in L83–94 as follows.
Medical-grade PAPRs are widely used as protective equipment against airborne in-fectious diseases. First, PAPRs are equipped with high-performance nonwoven fabric fil-ters such as HEPA filters [15], which can capture over 99.97% of particles at the most pen-etrating particle size (MPPS), typically around 0.3 μm [16, 17]. In general, there is a trade-off between aerosol filtration efficiency and airflow resistance in nonwoven filters; however, new types of filters based on novel principles that overcome this limitation have also been developed [18,19].
Second, PAPRs maintain positive pressure inside the hood, preventing aerosol infil-tration from the outside. As a result, the Assigned Protection Factor (APF) of hel-met/hood-type PAPRs reaches 1,000 [20,21], significantly surpassing that of half-mask respirators (APF = 10) and N95 masks. In contrast, N95 masks exhibit negative pressure during inhalation, structurally allowing outside air to enter through gaps.
- Table 1 - the flow rate [L/min] line contains values that are many times higher than the human breathing flow rate; studies typically assume air consumption of 6L/min; the maximum flow rate I have seen was 18L/min for an athlete with very high effort. Shouldn't it be L/h?
→ This may be unexpected. The explanation in L144–152 has been revised as follows.
The flow rates of the PAPRs shown in Table 1 range from 180 to 200 [L/min] for com-mercially available units, and 400 [L/min] for the prototype developed by the au-thors—values significantly higher than the resting respiratory flow rate of an adult male (ap-proximately 6 L/min) [30]. This high flow rate is intended to prevent the accumulation of exhaled carbon dioxide within the PAPR hood. With such high air supply rates, these PAPRs achieve relatively comfortable wearing conditions without fogging of the face shield due to exhaled moisture. However, such high flow rates can also be considered a substantial waste, suggesting that flow path design is critical to ensure the rapid expulsion of exhaled air from the hood.
- Line 128, “mass production” - Was the declared $40 calculated for mass production?
→ The amount of $40 is the total sum of the component procurement costs. The explanation in L126–137 has been revised as follows.
The prototype PAPR [25] developed in this study incorporates a HEPA filter (cut from a commercially available air purifier filter), a pump, and a battery, with a total component procurement cost of approximately USD 40. The prototype was assembled at Cebu Tech-nological University (Cebu city, Philippines), and the components were procured through online shopping sites (such as Lazada). As a result, the cost of components was kept low. The device supplies filtered air, purified via a HEPA filter capable of removing over 99.97% of 0.3 μm particles, into the hood to maintain a positive pressure environment. Consequently, even if small gaps exist between the hood and the user’s face, outside air inflow can be physically prevented. Exhaust is released passively through the exhaust fil-ter via differential pressure, which may offer some virus emission suppression if the wearer is infected. This design achieves the core functionality of a commercial medi-cal-grade PAPR.
- Line 133, “droplet” - for accuracy, the diameter range of pathogenic particles for which this assumption applies should be specified.
→ I believe this is an important point, but even among experts, opinions appear to be divided. In the early stages, particles larger than 5 μm were often classified as droplets, but more recently, some argue that droplets should be defined as particles larger than 100 μm. In this context, we have deliberately chosen to leave the definition ambiguous.
- Line 136, “with Rt_target = 0.8” - It would be very interesting to know what infection risk model was used to determine the reproduction rate.
→ I also believe this is an important point; however, in this paper, we adopt "0.8" as one example of a "sufficiently convergent value."
- Line 136, “the waring rate is 100%” - requires clarification of what this term means; does it mean that the entire population will use PAPR.
→ An explanation has been added as follows. (L159)
100% (i.e., the entire population wears the device at all times),
- Lines 156-158, “The air supply flow is changed by adjusting the speed of the variable-frequency centrifugal fan according to the air velocity at the half mask outlet monitored by the modular wind speed transmitter.” - this sentence contradicts itself. According to this sentence - The fan speed was controlled by air flow and simultaneously by air speed (not velocity, it is not the same). After all, it is the same, assuming that the physicochemical parameters of the air did not change significantly.
→ Indeed, the explanation (L179-181) was incorrect. It has been revised as follows:
The air supply flow is changed by adjusting the speed of the variable-frequency centrifugal fan according to the respiratory airflow states (exhalation or inhalation) detected at the half mask outlet.
- Line 161, “In the future” - Most systems currently in use already use this solution.
→ As shown in L182–194, the explanation in this section has been revised as follows. The technique of accurately detecting respiratory states (whether exhalation or inhalation) solely from differential pressure information—without inserting sensors near the nose—has not yet been implemented in PAPR systems. This is the key technical contribution that this paper aims to provide. Furthermore, we outline a future vision in which this technology leads to the realization of a “Wearing Rate Network Management System.” This system could enable the coexistence of two goals: efficient infection control by governments and the preservation of individual freedom to choose when and where to forgo PAPR usage.
Network connectivity and integration with information systems are also important for PAPRs. We have developed a PAPR equipped with a controller that includes Bluetooth connectivity [34]. In the future, advanced pressure control synchronized with breathing activity and integration with network systems for recording and verifying wearing rates may allow us to reconcile individual freedom with effective infection control [1,2].
In this study, we experimentally derive a fluid dynamic model—i.e. a calibration formula that expresses the flow rate Q of the supply and exhaust units as a function of pump duty and differential pressure ΔP—with the aim of enabling real-time pressure control synchronized with respiratory activity. This pressure control assists breathing by shifting the internal pressure inside the hood to a lower level during exhalation and to a higher level during inhalation, based on the detection of respiratory phases. Furthermore, we attempt to detect respiratory activity (i.e., exhalation and inhalation) based on the de-rived model.
- Line 173, “mimics human breathing” - what the piston displacement was, whether it was a uniform movement (rectangular course), or a variable sinusoidal or, real based on previously collected averaged measurements of the flow rate of a real person. This is of great importance for leak simulation, as shown by the results of research in already published articles. A sudden change in flow, causing a sudden change in the relative pressure under the mask, causes non-linear, inertial changes in the effective cross-section of the gaps, the degree of mask fit to the face.
→ The operation is performed manually, with the operator adjusting it arbitrarily (e.g., in sync with their own breathing). The second-to-last block at the end of Section 4.3 (L533-540) has been revised as follows.
As shown in Figures 5 and 6, the disturbance flow manually generated using the Respiratory Airflow Simulator corresponded to a level of physical activity between com-plete rest and moderate exercise, but closer to complete rest [32] in terms of respiratory rate and tidal volume. However, in terms of temporal variation in flow, the simulated pattern does not closely resemble actual respiratory patterns [36]. A future challenge will be to modify the Respiratory Airflow Simulator to use a linear motor to more accurately repro-duce time series data of actual respiratory flow. The introduction of an automatic artificial lung device could also be considered.
- Lines 203-204, “mimics the respiratory flow” - What was the flow rate, what was the length of each phase of the breathing cycle, was there a break between exhalation and inhalation? The standard cycle is 6L/min, 2.5s inhalation, 2.5s of exhalation and 1s break.
→L353–356 has been revised as follows.
In Figure 5, the four cycles of the disturbance flow Q3m span 13.9 seconds, from ap-proximately t = 2.3 s to t = 16.2 s. This corresponds to a respiratory cycle of 3.5 seconds. The average half-cycle flow volume (time-integrated, representing either exhalation or inhala-tion) is 5.6 liters. Therefore, the simulator operates at approximately 17.3 breaths per mi-nute with a tidal volume of 1.4 liters per breath.
- Line 237, “is driven manually” - I strongly encourage you to improve this method; why wasn't an artificial lung system used, as in most similar studies?
→ The following sentence has been added at L539-540.
The introduction of an automatic artificial lung device could also be considered.
- Line 248, “EPS32” - Is this really EPS32 or isn't it the ESP32 popular among young people?
→ I had made a mistake. It has been corrected.
- Line 254, “calculated from the regression equations” - requires an explanation of exactly what equations are meant.
→ The following sentence has been added to L282–283.
The regression equations for calculating Q1 and Q2 are shown the next section.
- Lines 259-260, “the dependence of the supply flow rate Q1 of the supply unit on the differential pressure ΔP and PWM duty value d” - so far it could be understood that the air intake fan was to be controlled by changing the PWM duty cycle?
→ Yes. The intake fan power (inhalation power) is varied by adjusting the PWM duty of the motor. Under conditions where dynamic effects can be neglected, the flow rate at a given power (PWM duty) and voltage depends solely on the differential pressure. Similarly, the exhaust flow rate also depends solely on the differential pressure when dynamic effects are negligible. These models are derived under static conditions. Figure 5 demonstrates whether these models remain valid under slightly dynamic conditions, such as when respiratory disturbance is introduced. Furthermore, Figure 6 illustrates whether the models also function when the PWM duty is dynamically varied..
- Figure 4 - It's hard to understand how it's possible that for zero flow there's a pressure difference?
→A fan for exhaust is installed in the pressure buffer to generate negative pressure. Additionally, by completely sealing the exhaust outlet, the maximum static pressure within the pressure buffer—achievable by the intake power of the supply unit—is realized using this experimental setup.
- Figure 5 - did Q1 and Q2 take on a value of about 200L/min? that is really a lot of air. How huge would the pressure drop have to be on the filter material, even with such a large surface area as shown earlier in Figure 3.
→ The flow rates used in the experiments are set to roughly match the specifications of the PAPR shown in Table 1. (Related to “Reply to comment-6”).

Round 2
Reviewer 3 Report
Comments and Suggestions for Authors
I have read the authors' explanations regarding my comments with great care. I must admit that the authors have responded to each comment and remark in detail and in full. I cannot fully agree with all of the authors' statements, but I do accept other points of view, for example, regarding the amount of air forced through the user's breathing zone or the method of breathing simulation. The authors have modified the content of the manuscript by adding explanations and specifying statements. I believe that the content of the manuscript has been significantly improved and in its current form facilitates a better understanding of the authors' idea regarding a powered air-purifying respirator.